# Cost–Utility Analysis of Dupilumab for the Treatment of Chronic Rhinosinusitis with Nasal Polyps (CRSwNP) in Italy

**DOI:** 10.3390/jpm12060951

**Published:** 2022-06-10

**Authors:** Eugenio De Corso, Gianluca Furneri, Daria Salsi, Francesca Fanelli, Gianluca Ronci, Giovanna Sala, Rossella Bitonti, Domenico Cuda

**Affiliations:** 1Otolaryngology Institute-Department of Head and Neck, Fondazione Policlinico Universitario A Gemelli IRCCS, 00168 Rome, Italy; eugenio.decorso@gmail.com; 2EBMA Consulting, 20077 Melegnano, Italy; gianluca.furneri@ebmaconsulting.com; 3Department of Otorhinolaryngology, G. da Saliceto Hospital, 29121 Piacenza, Italy; salsidaria@hotmail.com; 4Sanofi S.p.a., 20158 Milan, Italy; francesca.fanelli@sanofi.com (F.F.); gianluca.ronci@sanofi.com (G.R.); giovanna.sala@sanofi.com (G.S.); 5Director Department of Otorhinolaryngology, G. da Saliceto Hospital of Piacenza, 29121 Piacenza, Italy; d.cuda@ausl.pc.it

**Keywords:** dupilumab, cost–utility analysis, CRSwNP

## Abstract

The objective of this analysis was to estimate the incremental cost–utility ratio (ICUR) of dupilumab as an add-on treatment to best supportive care (BSC), versus BSC alone, in Italy for severe uncontrolled chronic rhinosinusitis with nasal polyps (CRSwNP). A simulation of outcomes and costs was undertaken using a 1-year decision tree, followed by a lifetime horizon Markov model. Clinical data were derived from a pooled analysis of two studies (SINUS-24 NCT02912468 and SINUS-52 NCT02898454). The Italian National Healthcare Service (NHS) perspective was considered. Model robustness was tested through sensitivity analyses. In the base-case analysis, treatment with dupilumab + BSC resulted in an increase in quality of life-adjusted survival (+1.02 quality-adjusted life years (QALY-gained)), compared to the BSC alone. The resulted ICUR was €21,817 per QALY-gained and it is below the acceptability threshold commonly used in Italy. Both one-way deterministic and probabilistic sensitivity analyses confirmed the robustness of base-case results. The cost–utility analysis showed that dupilumab, as an add-on treatment to BSC, is a cost-effective therapeutic alternative to BSC in the treatment of patients with severe uncontrolled chronic rhinosinusitis with nasal polyps, confirming that it is economically sustainable.

## 1. Introduction

Chronic rhinosinusitis with nasal polyps (CRSwNP) is an inflammatory condition of the nose and paranasal sinuses [1,2,3]. Nasal polyps are inflammatory lesions, typically bilateral, originating from the ethmoid sinus [3]. Symptoms associated with CRSwNP include nasal obstruction, reduction in sense of smell, nasal discharge, and sleep disturbances [4]. CRSwNP has a prevalence of 2–4% of the adult European population [5]. CRSwNP is often associated with respiratory diseases, such as asthma, aspirin sensitivity, and idiopathic bronchiectasis [5]. The underlying mechanisms that contribute to the chronic sinonasal inflammation observed in CRSwNP are not completely defined: it is hypothesized that an impaired sinonasal epithelial barrier could lead to increased exposures to inhaled pathogens, antigens, and particulates that, in the setting of a dysregulated host immune response, could promote chronic inflammation [3].

Despite being relatively easy to diagnose, CRSwNP is characterized by several unmet needs, such as poor knowledge of etiology and the association with some asthma phenotypes [6]. In addition, CRSwNP was found to adversely affect many aspects of a person’s quality of life (QoL), assessed through both the Study Short Form-6D (SF-6D) and the Euroqol 5-Dimension (EQ-5D) questionnaires, involving general-health survey inquiries measuring physical functioning, role limitations, social functioning, bodily pain, mental health, and vitality [7,8].

Furthermore, it can cause a substantial economic and patient burden. As a matter of fact, nasal obstruction is the main cause of discomfort in everyday life, associated with hyposmia/anosmia and sleep disturbances. In addition, patients have reported missed workdays (CRS is associated with an average of 4.8 days of missed work per year [9]) and missed leisure and family time due to the symptoms of CRSwNP [10,11,12,13]. In a recent study in Europe, direct costs for patients with CRSwNP were €1501 per patient/year, with indirect costs of €5659 per patient/year, largely due to outpatient/hospital visits and productivity losses, respectively [14].

Medical treatment options for patients with CRSwNP are limited and remain a challenge for the specialists. According to the most recent guidelines [15,16,17], both topical and oral corticosteroids are recommended as initial medical therapies for affected patients, for mild or severe disease, respectively. Existing data support the infrequent use of steroids in the immediate and short-term periods for patients with CRSwNP; however, their long-term benefits are limited [13,18,19]. Systemic corticosteroids treatment can lead to serious treatment-related adverse effects, and even short-term corticosteroid use is associated with an increased risk of acute complications, such as sepsis, venous thromboembolism, and fracture [13,18,19]. If medical therapies do not produce effects, surgery is an option for severely affected patients, followed by oral/intra-nasal corticosteroids, nasal saline irrigations, and long-term antibiotics [20]. Nevertheless, postoperative nasal polyp recurrence is common, with reported recurrence rates of approximately 40% of patients within 18 months of surgery, to nearly 80% within 12 years [21] From an economic point of view, the need for repeated surgical interventions and the side effects of oral steroids (osteoporosis, diabetes, cataracts, obesity, hypertension, and glaucoma), which are often taken for long periods, can cause an increase in healthcare expenses. Moreover, patients receiving rescue treatments (corticosteroids, antibiotics) or undergoing surgeries incur higher direct medical costs than patients who do not receive rescue treatments [22]. 

Despite the presence of conventional medical treatments, there is still an unmet need for patients with inadequately controlled disease. Dupilumab is a fully human Ig4 monoclonal antibody directed against the interleukin-4 receptor subunit α (IL-4Rα) of IL-4 and IL-13 receptors, blocking the effects of both cytokines, which are involved in T2 inflammation and, therefore, also in Nasal Polyps. Dupilumab is the first, approved biologic treatment indicated for severe CRSwNP. 

The European Medicines Agency (EMA) approved the use of dupilumab in 2019 for the treatment of adults with severe uncontrolled CRSwNP for whom therapy with systemic corticosteroids and/or surgery has not provided adequate disease control [23]. Currently, in Italy, dupilumab is reimbursed for the treatment of chronic rhinosinusitis with severe uncontrolled CRSwNP in addition to background therapy with intranasal corticosteroids in adults with a NPS score ≥ 5 or a SNOT-22 score ≥ 50 for whom treatment with systemic corticosteroids and/or surgery does not provide adequate control of the disease [24,25,26,27]. Two large, phase 3 trials, SINUS-24 and SINUS-52, have demonstrated that dupilumab is well-tolerated and significantly improves endoscopic, clinical, radiological, and patient-reported outcomes; a marked improvement in the sense of smell was also observed [28,29].

The objective of this study was to estimate the incremental cost–utility ratio (ICUR) of dupilumab compared to best supportive care (BSC) as an add-on treatment in patients with inadequately controlled CRSwNP. Dupilumab was compared to BSC, which includes: (i) nasal irrigation with saline solution and inhaled corticosteroids (INCS); (ii) oral corticosteroids (OCS); (iii) at least one surgery.

## 2. Materials and Methods 

### 2.1. Model Design

A cost–utility analysis was developed to compare costs and clinical outcomes associated with dupilumab administered subcutaneously 300 mg every two weeks plus BSC vs. BSC alone. The simulation of costs and outcomes was performed using a decision tree (with probabilistic nodes at 24 and 52 weeks, Figure 1a) linked to a Markov model, over a lifetime horizon, to estimate the long term of dupilumab and BSC (Figure 1b). After completing the simulation through the decision tree, the patients entered the Markov model.

As shown in Figure 1a, all patients with CRSwNP enter the model at time t_0_ and receive: (i) dupilumab + best supportive care (referred to as “dupilumab”); or (ii) best supportive care (not in combination, referred to as “BSC”). At the first assessment point of Week 24, a clinical check is performed to determine the response in both arms. Responders will remain in the decision tree until Week 52.

Based on the assessment results at the end of the 1-year decision tree, responders enter into the Markov model, in the health state called “Controlled disease” (Figure 1b). At each assessment point in the decision tree, non-responders discontinue active treatment and move either to the “Inadequately controlled disease”, or “Surgery” (Figure 1b). During the Markov-modelled life-time observation, patients in the “Controlled disease” health state can remain in the same health state until discontinuation, due to lack of efficacy/treatment intolerance. After discontinuation, these patients switch to the health state “Inadequately controlled disease” (Figure 1b). 

Patients in the “Inadequately controlled disease” state either stay in that state or move to the “Surgery” state, in which they undergo a surgery procedure. After surgery, patients move to the “Post-operative controlled” health state. Patients in the “Post-operative controlled” state can either stay in that health state or move to the “Post-operative uncontrolled” state if their disease condition becomes uncontrolled. Patients in the “Post-operative uncontrolled” state may move to the “Surgery” state for a revision surgery. Patients can have multiple surgeries in the model. It is possible to die in any of the health states (move to the “Death” state). 

A discount rate of 3.0% was applied to both costs and effects [30]. In the base-case, the incremental cost–utility ratio (ICUR) of dupilumab compared to BSC was estimated adopting the National Healthcare Service (NHS) payer perspective and considering only direct healthcare resources reimbursed and funded by the NHS.

### 2.2. Clinical Inputs

#### 2.2.1. Baseline Characteristics

Baseline characteristics of patients entering the model were extracted from the intention-to-treat (ITT) population of the pooled analysis of SINUS-24 and SINUS-52 trials [28,29]. The mean age among patients was 51.39 years, and 60.4% of patients were male.

#### 2.2.2. Response at 24 and 52 Weeks

The primary endpoints of the SINUS-24 and SINUS-52 studies evaluated the change from baseline in the NPS score (Nasal Polyp Score) measured by nasal endoscopy from baseline and the change in the NCS score (Nasal Congestion Score) based on patient assessment from baseline through weeks 24 and 52, respectively. In this analysis, the efficacy of the alternative treatments was defined in terms of therapeutic response, through the combined endpoint of improvement of at least 8.9 points in the score of the 22-item sinonasal outcome test (SNOT-22 test) and improvement of at least 1 point in the nasal polyposis score (NPS), compared to the scores at baseline (hereinafter defined as SNOT-22 CFB (change from baseline) ≥ 8.9, and NPS CFB ≥ 1). Figure 2 shows the response rates at 24 and 52 weeks for the two alternatives [28,29,31].

#### 2.2.3. Probability of Sustained Response for Years 2–5+

The long-term response to the treatment alternatives was assessed by a panel of five clinical experts [32]. The survey asked the physicians to provide estimates of the probability of sustaining the Year 1 response (and, consequentially, quality-of-life benefits) in subsequent years for dupilumab and BSC. The responses from the physicians were averaged and the values from the base-case assumptions for long-term sustained response were obtained. Panelists estimated a 98% probability of sustained response in both treatment arms every year [33].

#### 2.2.4. Clinical Data Inputs for Patients in “Inadequately Controlled Disease” Health State and Post-Operative Period

Patients with inadequately controlled disease can remain in the same health state or change to the “Surgery” health state. According to the results of the pooled analysis of the SINUS-24 and SINUS-52 studies [28,29,31], 15.1% of patients are not eligible for surgery. These patients will remain in the “Inadequately controlled disease” health state. For the remaining 84.9% of patients (eligible for surgical treatment), it was assumed that 1.9% annually need surgical retreatment and move to the “Surgery” health state [34]. Finally, it was assumed that surgery can be performed only on patients under 70 years old.

Table 1 shows the clinical inputs used for the “Inadequately controlled disease” and “Post-operative” health states. In order to reflect the clinical practice in the model, it was assumed that 40% of patients switch to the “Post-operative uncontrolled” health state from the “Post-operative controlled” state.

### 2.3. Utility Inputs

Utility weights, used in the model to calculate quality-adjusted life years, were derived from the pooled analysis of the SINUS-24 and SINUS-52 studies [28,29,31]. Utility weights were estimated using the crosswalk method described in the Chevalier et al. (2010) [35] and van Hout et al. (2012) [36] studies, using English tariffs. Two sets of utilities were used in the model (Table 2): (a) utilities in the decision tree (Year 1 of the simulation); (b) utilities in the Markov model (Year 2 and following of the simulation). Utility weights were assumed to depend only on the health state and not on received treatment, except for utility weights for week 12–24. During expert validation, this assumption was considered conservative for dupilumab, as it would not fully capture the benefit of receiving a disease-modifying treatment.

### 2.4. Cost Inputs

Consistently with the perspective adopted in the base-case, the following direct healthcare costs were identified, measured and quantified: (i) drug acquisition costs; (ii) disease management costs; (iii) adverse events costs; (iv) surgery costs. 

To calculate the annual acquisition cost of treatment with dupilumab, the unit cost of a vial (ex-factory price per vial, published in the Official Gazette, net of any discounts applied to public NHS structures) [24,25,26] was multiplied by the number of administrations per year. The dosage of dupilumab was derived from the Summary of Product Characteristics [23], the administered dose of dupilumab is 300 mg every 2 weeks. In the base-case, a compliance rate of 100% was used. An alternative scenario analysis, with a lower compliance rate of 96.95%, derived from the RJL study of van der Lans et al. (2021) [38], was performed.

Acquisition cost of treatment with BSC was assumed to be €0, presuming that the costs of these therapies are, overall, minimal. Administration costs were assumed to be equal to €0, since dupilumab is administered subcutaneously and almost exclusively in a home setting. Furthermore, for the same reason, it was assumed a zero cost of administration training for dupilumab; the study of Bhattacharyya et al. (2019) was used to determine the costs associated with the management of CRSwNP, based on therapeutic response (Table 3) [22]. In Italy, it was assumed that the lack of disease control mostly leads to the use of oral corticosteroid therapy or endoscopic surgery. Costs derived from the study of Bhattacharyya et al. (2019) [22] were converted into EUR (IT) using the purchasing power parities (PPP) 2021 (coefficient 0.648) [39]. 

The costs associated with commonly observed adverse events from the SINUS-24 and SINUS-52 trials were calculated by multiplying the unit costs of adverse events in Italian practice by the respective incidence rate (Table 4).

Finally, the costs of surgery were estimated taking into account the costs of computed tomography (€104, derived from the average of the tariffs of codes 87.03.2 and 87.03.3 [40]), the cost of a generic surgery (€2009, tariff of day hospital code 053 “Surgeries on paranasal sinuses and mastoid, age > 17 years” [43]), and annual follow-up costs after surgery (€103, calculated assuming *N* = five specialist visits per year [33]).

### 2.5. Sensitivity Analysis

One-way sensitivity analysis (OWSA) and probabilistic sensitivity analysis (PSA) were performed to identify model parameters with the largest effect on ICUR, and to evaluate the overall robustness of the base-case analysis. For the deterministic sensitivity analysis, the baseline value of each parameter was modified to the upper and lower limits of its 95% confidence interval (95% CI). If the CI was not available, a variation of ±10% from the baseline value was used.

For the probabilistic analysis, the following probability distributions were used: (i) Gamma for costs and resource use; (ii) Beta for the probabilities/proportions, discontinuation rates, and incidences of adverse events; (iii) Normal for utilities.

### 2.6. Scenario Analyses

In addition to the base-case analysis (from the NHS perspective) two alternative scenario analyses were conducted: (i) adopting the societal perspective; (ii) including a lower compliance rate, closer to the real clinical practice. In the first scenario analysis, the costs related to the productivity loss of the patients were considered. These costs were calculated considering the value of one hour of productivity loss (€/h 29.80) [44], the number of working hours in a day (8 h) and the number of working days lost. In the second scenario analysis, a compliance rate of 96.95%, derived from the RJL study of van der Lans et al. (2021) [38], was used.

### 2.7. Model Validation

The model was subjected to a thorough validation process in accordance with guidelines for validation put forth by the International Society for Pharmacoeconomics and Outcomes Research and the Society for Medical Decision Making’s Modelling Good Research Practices Joint Task Force [45]. Face validity was tested throughout the model development with external experts. Internal validity was tested as part of a rigorous quality-control process, during which the researchers not involved in model development checked the accuracy of all of the data extracted from the literature, the logical structure of the model, and the accuracy of all of the calculations and programming. External validation was not possible because this is the first cost-effectiveness model for long-term treatment with a biologic in CRSwNP.

## 3. Results

### 3.1. Base-Case Analysis

In the base-case analysis (Table 5), dupilumab was more effective, in terms of quality-of-life-adjusted survival (+1.02 quality-adjusted life years (QALY-gained)) compared to BSC. The introduction of dupilumab leads to an increase in drug treatment costs (+€76,383), which are partially offset by a decrease in disease management costs (−€53,850), and by an improvement in quality of life. The resulted ICUR was €21,817 per QALY-gained.

### 3.2. Sensitivity Analysis

Both one-way deterministic (OWSA) and probabilistic sensitivity analysis (PSA) confirmed the robustness and reliability of the base-case results. The results of the deterministic sensitivity analysis are shown in the tornado graph (Figure 3). A slight variability in the ICUR was estimated (ICUR min €14,502, ICUR max €30,292).

The results of the probabilistic sensitivity analysis (PSA) are reported in Figure 4 (acceptability curve of cost–utility) and in Figure 5. The acceptability curve of the cost–utility analysis showed that, considering the informal range of acceptability proposed for Italy (€25,000–€40,000 per QALY-gained) [46] for the willingness to pay (WTP), dupilumab would have a 78.3% probability of being cost-effective compared to best supportive care (the midpoint of the proposed acceptability range was considered).

### 3.3. Alternative Scenarios

The results of the alternative scenario analysis 1 (adopting the social perspective) confirmed the results of the base-case (Table 6). Dupilumab generated a small increase in direct costs and a slight decrease in indirect costs. This saving in indirect costs is associated with the lower productivity loss generated by the introduction of dupilumab. The resulting ICUR was €21,503.

Furthermore, the results of the alternative scenario analysis 2 (considering a lower compliance rate) confirmed the results of the base-case. The resulting incremental cost–utility ratio was €19,536.

## 4. Discussion

The objective of this analysis was to compare, in terms of costs and benefits, dupilumab vs. the best supportive care (BSC), as an add-on treatment in patients with inadequately controlled CRSwNP, from the Italian NHS perspective. In addition to the treatment of severe uncontrolled CRSwNP, dupilumab is also reimbursed for the treatment of severe asthma (with type 2 inflammation) in adults and adolescents [24,25,26,27]. Such pathologies often coexist in the same patient [5]. As a matter of fact, they are characterized by a common etiopathogenetic mechanism, namely the activation of type 2 inflammation mediated by several interleukins, including IL-4, IL-13, and IL-5. The biologic drugs currently available (anti-IgE, anti-IL4Rα, anti-IL5, and anti-IL5Rα) for severe asthma are currently undergoing clinical development studies to extend their availability to patients with nasal polyposis.

The introduction of biologic drugs and their expanding use over the past few years have transformed the management of patients suffering from severe atopic diseases. Unfortunately, their higher price also creates new challenges in terms of access and sustainability [47]. However, dupilumab proved to be a new therapy with an affordable increase in terms of costs compared to the current standard of care. Furthermore, the clinical efficacy associated with dupilumab, compared to the standard of care, was high [28,29]. 

A drug such as dupilumab, indicated for the treatment of various pathologies, could lead to significant savings for the NHS, as demonstrated by the work of Jommi et al. (2020) [48]. Dupilumab was also shown to be a cost-effective alternative for the treatment of patients eligible for treatment in Italy with atopic dermatitis or asthma [49,50,51]. Before the introduction of biologics, therapies for CRSwNP were designed to decrease inflammation or weaken the factors that may be causing inflammation. These therapies were inexpensive and had good evidence to support their use. When medical therapy does not resolve a patient’s symptoms, functional endoscopic sinus surgery (FESS) can be used. Biological therapies represent a potential “game changer” in the treatment of CRSwNP with benefits of improving quality of life assessments, reducing polyp scores, and reducing nasal congestion scores, yet there is a lack of durability with discontinuation of the drug. In addition, these therapies are often expensive [52]. 

In this context, the assessment of the cost–utility of new therapies is crucial to compare the costs and outcomes introduced with the availability of a new treatment. In Italy, public decision-makers have not defined acceptability thresholds for ICURs; however, in other European countries, the threshold is generally placed around €50,000/€60,000 per year of life gained [53,54]. 

In addition, the Italian Society of Health Economics (AIES) [46] has proposed an informal range of acceptability for Italy equal to €25,000–€40,000 per QALY-gained. The present cost–utility analysis has shown that, using the ex-factory price proposed to the Italian NHS, net of hidden discounts, dupilumab is a cost-effective alternative to BSC (with an ICUR falling both within the proposed range from the AIES and below the €50,000/QALY-gained threshold), for the treatment of severe CRSwNP. An additional confirmation of the robustness of the base case analysis is represented by the scenario analysis. The analysis from the perspective of society was conducted to include indirect costs that are not reimbursed by the health system, but which impact the lives of patients. It was estimated that the introduction of dupilumab in the Italian context reduces the costs associated with loss of productivity compared to BSC, thanks to the greater therapeutic efficacy associated with dupilumab.

In the context of any economic evaluation, a critical assessment of the methodology adopted and of the potential limitations of the study is essential. In our opinion, the main methodological aspects of the present study that require in-depth analysis are: (i) the use of an expert panel to estimate the long-term efficacy of dupilumab; (ii) the use of the results of the study of Bhattacharyya et al. (2019) [22] to estimate the costs of managing the disease; (iii) the use of conservative assumptions in setting up the cost–utility analysis. 

In regard to the first point, it is important to highlight that data estimation through expert opinion is not the most robust approach. However, in the absence of alternative data, this methodology was the only one possible for conducting the analysis. Furthermore, the same source was already validated in the evaluation of dupilumab for the treatment of adult patients with moderate to severe AD, performed by the National Institute for Health and Care Excellence (NICE) [55]. 

Regarding the second point, the study of Bhattacharyya et al. (2019) [22], used to determine disease management costs, may not be exhaustive in identifying all of the cost items associated with CRSwNP. It is an US study; therefore, some cost items may not be relevant for the Italian context. In addition, it was hypothesized that non-responder patients are treated with oral corticosteroids and FESS. However, we did not consider asthma patients in this group, who are generally less responsive to treatment, and this may have a greater impact on costs. Nevertheless, these are conservative hypotheses; the inclusion of different cost items could generate an increase in disease management costs that could create an advantage for dupilumab. 

Regarding all of the economic inputs, a predominantly conservative approach was adopted, such as excluding some costs (e.g., acquisition costs of the BSC) that could favor dupilumab. The improvements of some specific comorbidities associated with CRSwNP, such as otitis and asthma [20], during treatment with dupilumab were not considered; in addition, the costs of managing adverse events related to surgery were not considered, as they can only be derived from clinical practice (not present in clinical studies). Finally, the impact of the treatments on the patients’ olfactory dysfunction was not considered in the analysis. Treatment with dupilumab significantly improves patients’ sense of smell [56]. Although this aspect does not impact the cost of disease management, its role in improving the quality of life in patients is significant compared to other available therapeutic alternatives.

It should be highlighted that some of the simplifying assumptions mentioned above may have generated accuracy errors, although of minimal relevance for the purposes of calculating the ICUR. However, the uncertainty was tested through sensitivity analyses which confirmed the robustness of the results. Clearly, an update of the results of this analysis is recommended when real-world data from the use of dupilumab becomes available. 

Despite some methodological limitations, typical of the economic models developed immediately after the conclusion of clinical trials, we believe that the analysis is to be considered informative for prescribers and policy makers.

## 5. Conclusions

Our cost–utility analysis showed that dupilumab, as an add-on treatment to BSC, is an effective therapeutic alternative to the BSC in the treatment of patients with CRSwNP. The resulting ICUR (base-case: €21,817/QALY-gained) is favorable and below the acceptability thresholds commonly used in Italy. Both sensitivity analyses confirm that the therapeutic approach with dupilumab is “economically appropriate” and sustainable.

## Figures and Tables

**Figure 1 jpm-12-00951-f001:**
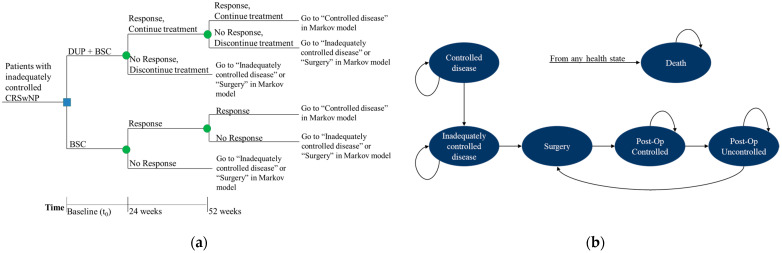
Model structure: (**a**) Decision tree; (**b**) Markov model. Note: BSC, best supportive care; CRSwNP, chronic rhinosinusitis with nasal polyps; DUP, dupilumab.

**Figure 2 jpm-12-00951-f002:**
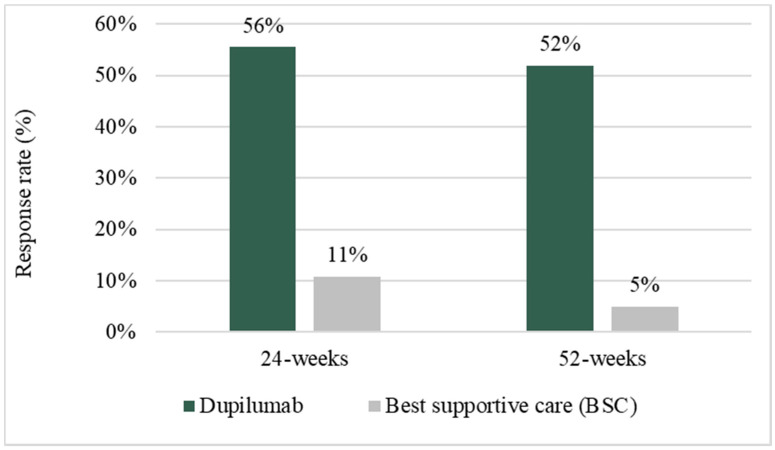
Response rate at 24 and 52 weeks.

**Figure 3 jpm-12-00951-f003:**
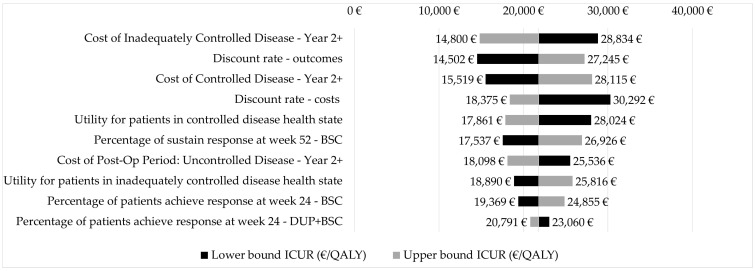
Tornado graph (deterministic sensitivity analysis): dupilumab + BSC vs. BSC (base-case). Note: BSC, best supportive care; DUP, dupilumab; ICUR, incremental cost–utility ratio; QALY, quality adjusted life year.

**Figure 4 jpm-12-00951-f004:**
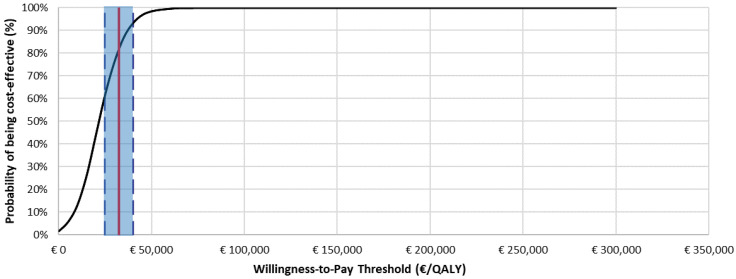
Cost–utility acceptability curve: dupilumab + BSC vs. BSC (base-case; *N* = 1000 simulations; ICUR in base-case: €21,817/QALY-gained).

**Figure 5 jpm-12-00951-f005:**
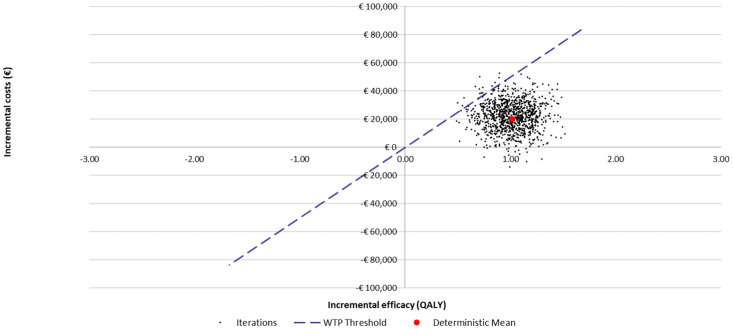
Results of the probabilistic sensitivity analysis.

**Table 1 jpm-12-00951-t001:** Clinical inputs for the “Inadequately controlled disease” and “Post-operative” health states.

Parameters	Value	Source
Clinical input for “Inadequately Controlled Disease” state		
Percentage of patients who are ineligible for surgery (%)	15.1%	Pooled analysis of the SINUS-24 and SINUS-52 studies [28,29,31]
Maximum age eligible for surgery (years)	70.0	Assumption
Annual transition probability of eligible patients moving to “Surgery” health state (%)	1.9%	PMSI, 2019 [34]
Annual transition probability of eligible patients moving to “Surgery” health state (%)	98.1%	Calculated
Clinical input for Post-operative period		
Annual transition probability of moving to “Post-operative controlled” health state from “Surgery” health state (%)	100.0%	Pooled analysis of the SINUS-24 and SINUS-52 studies [28,29,31]
Annual transition probability of remaining in the “Post-operative controlled” health state (%)	60.0%	Assumption
Annual transition probability of moving to “Post-operative uncontrolled” health state from “Post-operative controlled” (%)	40.0%	Assumption
Annual transition probability of moving to “Surgery” health state from “Post-operative uncontrolled” health state	1.9%	PMSI, 2019 [34]
Annual transition probability of remaining in the “Post-operative uncontrolled” health state (%)	98.1%	Calculated

**Table 2 jpm-12-00951-t002:** Utility data included in the model.

Utility	DUP	BSC	Description
Decision tree			
Week 0–12, baseline	0.769	0.769	Baseline utility from pooled analysis SINUS-24 and SINUS-52 studies [28,29,31]
Week 12–24, regardless of response	0.875	0.810	Treatment specific utility at 24 weeks [28,29,31]
After week 25, patient responders	0.891	0.891	Utility at 24 weeks, all treatments
After week 25, patient non-responders	0.808	0.808	Utility at 24 weeks, non-responder patients [28,29,31]
Markov model			
Controlled disease	0.913	0.913	Utility at 52 weeks, responder patients [28,29,31]
Inadequately controlled disease	0.776	0.776	Utility at 52 weeks, non-responder patients [28,29,31]
Surgery	0.820	0.820	Estimated based on utility gain from surgery with inclusion of short-term disutility [37]
Post-operative controlled disease	0.827	0.827	Assumed to be 0.051 higher than the utility of “Inadequately controlled disease” (0.776) [37]
Post-operative uncontrolled disease	0.760	0.760	Assumed equal to baseline utility of patients with two previous surgeries

Note: DUP, dupilumab; BSC, best supportive care.

**Table 3 jpm-12-00951-t003:** Disease management costs.

Decision Tree				
Response state	Therapy costs (€)		Medical costs (€)	
	Weeks 0–24	Weeks 25–52	Weeks 0–24	Weeks 25–52
Responder patients	€780	€910	€3345	€3903
Non-responder patients	€842	€982	€6790	€7922
Markov model				
Health state	Total costs (€)			
	Year 1		Year 2+	
Controlled disease	€8937		€8937	
Inadequately controlled disease	€16,536		€16,536	
Post-operative controlled	€8937		€8937	
Post-operative uncontrolled	€16,536		€16,536	

**Table 4 jpm-12-00951-t004:** Adverse events’ incidence rates and costs.

Adverse Event	Incidence Rate (N/Year) [28,29,31]		Unit Costs (€)	Source of Costs
	Dupilumab	BSC		
Injection site reaction	0.395	0.000	20.66	Code 89.7 [40]
Nasopharyngitis	0.275	0.287	16.31	Garattini et al. inflated to January 2022 [41]
Epistaxis	0.106	0.114	16.31	Garattini et al. inflated to January 2022 [41]
Headache	0.092	0.181	20.66	Code 89.7 [40]
Asthma	0.058	0.173	326.48	Access to the emergency room, inflated to January 2022 [42]

Note: BSC, best supportive care.

**Table 5 jpm-12-00951-t005:** Results of the cost–utility analysis.

Results of the Cost-Utility Analysis	Dupilumab + BSC (A)	BSC (B)	Difference (A−B)
Outcome			
Years of life adjusted for quality (QALYs)	17.15	16.13	1.02
Direct costs (€)			
Drug acquisition costs	€76,383	€0	€76,383
Disease management costs *	€275,517	€329,367	−€53,850
Adverse events costs	€1083	€1333	−€249
TOTAL direct costs	€352,983	€330,700	€22,283
Incremental cost–utility ratio (ICUR)			
ICUR (dupilumab vs. BSC) (€/QALY)	€21,817		

Note: * They include the costs of surgery. BSC: best supportive care; ICUR: incremental cost–utility ratio; QALY: quality adjusted life year.

**Table 6 jpm-12-00951-t006:** Results of the scenario analysis 1.

Results of the Cost-Utility Analysis	Dupilumab + BSC (A)	BSC (B)	Difference (A−B)
Outcome			
Years of life adjusted for quality (QALYs)	17.15	16.13	1.02
Costs (€)			
Direct costs	€352,983	€330,700	€22,283
Indirect costs	€163	€483	−€320
Total costs	€353,146	€331,183	€21,963
Incremental cost–utility ratio (ICUR)			
ICUR (dupilumab vs. BSC) (€/QALY)	€21,503		

Note: BSC, best supportive care; ICUR, incremental cost–utility ratio; QALY, quality adjusted life year.

## Data Availability

This study is an economic evaluation that involves decision analytic modeling and stochastic methods. All parameter values and assumptions used to support the model, which are mostly derived from published literature or expert opinion, are listed in the paper. The Excel-based decision analytic model is available on request.

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
