# Peer review of "Cost–Utility Analysis of Dupilumab for the Treatment of Chronic Rhinosinusitis with Nasal Polyps (CRSwNP) in Italy"

_jpm, 2022, doi:10.3390/jpm12060951_

Round 1

Reviewer 1 Report

The objective of this cost-utility analysis was to estimate the ICUR of dupilumab (300 mg SC every two weeks) as an add-on therapy to BSC, as compared with BSC alone, for the treatment of inadequately controlled CRSwNP in Italy. From simulations and models of clinical data derived from two studies, researchers showed that treatment with dupilumab added to BSC resulted in +1.02 QALY gained and the resulted ICUR was €21,817 per QALY gained. Given that the resulted ICUR was below the acceptability thresholds commonly used in Italy, the researchers concluded that this cost-utility analysis demonstrated that dupilumab is a cost-effective therapeutic alternative to BSC for the treatment of patients with severe uncontrolled CRSwNP.

The reviewer offers the following comments, by section of the manuscript, for the authors’ consideration.

Abstract

In the last sentence of the Abstract, the authors concluded that this analysis showed that dupilumab is a cost-effective “therapeutic alternative” to BSC for the treatment of severe uncontrolled CRSwNP in Italy. Considering that the stated objective of this analysis was to estimate the cost-utility of dupilumab as an “add-on therapy” to BSC, would a more accurate conclusion be that dupilumab added to BSC is a cost-effective alternative to BSC alone for the treatment of severe uncontrolled CRSwNP in Italy? In other words, as currently stated, the authors implied that dupilumab alone was cost-effective compared with BSC alone, which was not the comparative analysis of this study. Please consider revising. Perhaps an explanation that the “dupilumab” group was composed of dupilumab therapy plus BSC would suffice, if that was the intention of the conclusion.

Introduction

The authors indicated that CRSwNP can cause a substantial economic and patient burden (page 2, lines 50-53) and they provided examples of these burdens (e.g., discomfort, sleep disturbances, missed workdays). Notwithstanding these examples, the claim about the economic burden could be strengthened by providing cost data (e.g., indirect costs associated with missed workdays), if feasible.

Similarly, in describing the medical treatment options for patients with CRSwNP (page 2, lines 54-64), providing economic data (e.g., direct medical costs of treatments, indirect costs associated with side effects) could further substantiate the claims about healthcare expenses associated with managing CRSwNP. 

Additionally, providing efficacy and/or safety data for guideline-recommended medical treatments (e.g., corticosteroids, surgeries) could improve how the unmet need is framed. For example, how well do corticosteroids control CRSwNP? How effective are various types of surgeries in the short- and/or long-term management of the condition? The bottom line is that the readers would benefit from better understanding the prevalence of “inadequately controlled” CRSwNP.

Much of the information provided in the second to last paragraph of the Introduction (page 2, lines 70-89) is superfluous, namely the information about non-CRSwNP indications for dupilumab (lines 70-82). Consider omitting this information and focusing on just the CRSwNP indication for dupilumab.

Materials and Methods

In the third paragraph of the Model Design section (page 3, line 114-117), the researchers stated that dupilumab responders with confirmed response at Week 52 moved from the decision tree to the Markov model and entered in the health status called “Controlled disease” yet all remaining patients (including those in the BSC group) entered in the health status called “Inadequately controlled disease.” This description of the BSC group in the manuscript copy appears to disagree with Figure 1. Specifically, Figure 1a indicated that BSC group responders, like their dupilumab group counterparts, entered the Markov model in the health status called “Controlled disease” rather than “Inadequately controlled disease.” Please provide clarification of how the BSC group – responders and non-responders – were handled in the model design.

In the Clinical Inputs section, is Table 1 necessary? The data in this table are exactly described in the manuscript body (page 4, line 140).

Results

Because the reported range of probability proposed for Italy was 25,000 to 40,000 per QALY gained and the demonstrated ICUR for this analysis was 21,817 per QALY gained, it might be worth adding some shading to Figure 4 to demarcate the range of probability, as a point of reference for the reader.

Discussion

The second paragraph of the Discussion section (page 11) is very lengthy and, as such, might benefit from breakage into two paragraphs.

Conclusion

Please refer to the reviewer’s comment about the Abstract for potential modification to the first sentence of the Conclusion section (page 12, lines 391-392), particularly about dupilumab being a “therapeutic alternative” to BSC.

Reviewer 2 Report

               The authors present a study examining the cost-benefit of including dulipumab along with the standard of care vs the standard of care alone for the treatment of chronic rhinosinusitis with nasal polyps (CRSwNP). The work utilized a decision tree and Markov model to estimate an incremental cost-utility ratio based on two phase 3 clinical trials and previously reported costs associated with the management of CRSwNP. Using this approach, the authors show a +1.02 QALY difference in the dulipumab + standard of care compared to the standard of care alone. Furthermore, the estimated cost-utility ratio was 21,817 euro or 21,817/QALY. Overall this is a interesting study and while it could benefit from real-world data validation for conclusion purposes, it is a good contribution to the currently available literature.

Comments:

               -Figure 1’s text is fairly small and hard to interpret, authors should consider enlarging the text.

               -Authors should explain in more detail how the assumptions surrounding the probabilities of post-operative controlled remaining in health state and post-operative controlled moving to post-operative uncontrolled were made for the Post-operative period clinical inputs.

               -Overall, more model validation should be at least discussed. While the authors employed sensitivity analysis to ensure robustness of the results, more robust methods such as external validation should be included or the reason for their exclusion should be discussed.

               -Authors should explain in more detail why the utility weights are the same for both study cohorts as well as the impact these weights have on the overall model.

               -Line 355: “As regards to…” should be changed to “In regards to…”
